# Feasibility of a Full-Field Measurements-Based Protocol for the Biomechanical Study of a Lumbar Belt: A Case Study

Rébecca Bonnaire [1,2,3,*], Woo Suck Han [2], Reynald Convert [3], Paul Calmels [4,5] and Jérôme Molimard [2,*]

1   Institut Clément Ader (ICA), Université de Toulouse, CNRS, IMT Mines Albi, INSA, ISAE-SUPAERO, UPS, Campus Jarlard, F-81013 Albi, France
2   Mines Saint-Etienne, Univ Lyon, Univ Jean Monnet, INSERM, U 1059 Sainbiose, Centre CIS, F-42023 Saint-Etienne, France; han@emse.fr
3   Thuasne, 92300 Levallois-Perret, France; reynald.convert@thuasne.fr
4   Univ Lyon, UJM-Saint-Etienne, Laboratoire Interuniversitaire de Biologie de la Motricité, EA 7424, F-42023 Saint-Etienne, France; paul.calmels@chu-st-etienne.fr
5   Service Médecine Physique et Réadaptation, CHU Saint-Etienne, F-42055 Saint-Etienne, France
*   Correspondence: rebecca.bonnaire@mines-albi.fr (R.B.); jerome.molimard@mines-stetienne.fr (J.M.); Tel.: +33-477426648 (J.M.)

**Abstract:** Low back pain represents a major economic and societal challenge due to its high prevalence. Lumbar orthoses are one of the recommended treatments. Even if previous results showed their clinical effects, the detailed mode of action is still poorly known, making the device design difficult. A renewed instrumentation and experimental protocol should bring better insight into the lumbar brace–trunk mechanical interaction. This instrumentation should give detailed information on the basic physical or geometrical parameters: the pressure applied on the trunk, the body shape and the strain in the belt. The principal objective of this study was to propose and validate a new measurement protocol, based on pressure mapping systems and full-field shape and strain measurement. The feasibility of the protocol was tested along with its validity and repeatability. The influence of various parameters, which could cause changes in the measurements, was tested with six different belt configurations on one subject. Measurements were also performed to study the impact of posture on pressure and strain. Both pressure and strain appeared to be asymmetric from left to right. The pressure applied by the lumbar belt on the back varies with breathing and with posture. This study showed that full-field measurements were necessary to render the high variability of pressure or strain around the trunk, under recommendations of their use to guarantee a satisfying repeatability.

**Keywords:** low back pain; lumbar belt; full-field measurement; DIC; fringe projection; pressure mapping system



## 1. Introduction

Low back pain is defined as pain in the lumbar region of the body. In France, the prevalence of low back pain can be estimated at 50% per year and 7% to 8% of the population aged 30–64 years have disabling low back pain [1]. Low back pain represents a major economic and societal challenge due to the large number of persons requiring treatment for this pathology every year and the sick leave which it causes [1,2].

Non-specific low back pain has a mechanical origin and should be differentiated from so-called symptomatic forms, which are caused by trauma, inflammation, infection or a tumour. Three evolving typologies of low back pain can be differentiated according to the duration of their evolution [3]. Here, we focus on the subacute to chronic non-specific low back pain, i.e., pain lasting from several days to three months.

Treatment of chronic or sub-chronic non-specific low back pain is varied and has been poorly codified. It depends on numerous factors, including the cause of the pain, but also on the decisions of the subject's general practitioner or the behaviour of the subject himself [4].

An analysis of the literature reveals the diversity of treatments: drug treatments, surgery, physical and rehabilitation therapies, vertebral manipulation techniques, lumbar orthoses as belts or corsets. A non-specific low back pain symptom does not necessarily arise from a single lesion; it is, in contrast, multifactorial. For this reason, therapeutic recommendations are a multidisciplinary approach for the treatment (physician, psychologist, physiotherapist, social worker, etc.) and regular medical follow-ups [5–7].

Lumbar orthoses are one of the recommended treatments for low back pain. These orthoses are medical appliances or devices and can be standard models or made-to-measure. It has been demonstrated that lumbar orthoses produce postural correction by modifying lumbar lordosis [8–11]. They mechanically reduce the pressure on the intervertebral discs [12], but clinical results are variable [13–15]. They restrict the amplitudes of segmental and global mobility [16,17]. Along with these mechanical effects, it has been demonstrated that lumbar orthoses have many clinical and physiological actions. They have effects on pain and inflammation [18–20], on neuromuscular and proprioceptive activity and also have a muscle relaxant effect [21,22], with a functional benefit by facilitating the activities of daily life [19,23,24]. Then, the consumption of medicines is reduced, thereby reducing possible iatrogenic complications [23]. Many authors showed that lumbar orthoses do not negatively affect muscle strength [21,25–28].

However, it is worth mentioning that there are few studies to date concerning the use of lumbar belts and the studies that exist involve heterogeneous populations, belts from different suppliers and with varied designs and a variety of methodological study factors. Moreover, they are often combined with other types of treatment which can make the analysis difficult [16,24]. Finally, even if previous results showed their clinical effects, the detailed mode of action of lumbar belts is still missing.

Even though other phenomena could be envisaged, such as thermal or proprioceptive effects, the claimed mode of action is a pure mechanical effect. It consists in the compression of all of the lower trunk area. This would lead to a load transfer from the spine—and in particular the lumbar discs—toward the abdomen and to a change in the patient posture that would result in a delordosing effect. Due to these two reasons, the nociceptors present in the back of the intervertebral discs should be less loaded, and pain would be released. This mode of action is closely related to the pressure applied by the belt on the trunk, as shown, for example, in [12]. The Laplace Law is a simple way to describe the pressure generation of a compression textile on a body part [29]. It states:

$$P = \frac{T}{R} = E\frac{\epsilon}{R}, R \geq 0,\tag{1}$$

were $P$ is the pressure, $R$ the local curvature radius of the body part and $T$ the tension on the textile, that is the product of the textile stretch $\varepsilon$ and its stiffness $E$. Surprisingly, until now, no work has investigated the pressure distribution around the patient's trunk, even though this distribution is closely related to the belt design. For more detail, a renewed instrumentation and experimental protocol is necessary. This instrumentation should give detailed information on the basic physical or geometrical parameters, namely the pressure applied on the trunk, the body shape and the strain on the belt.

Following the Laplace Law, the applied pressure varies depending on the morphological patient characteristics, among others, his/her body shape and his/her posture. Thus, it is of great importance to have measurements of pressure and deformation fields all around the trunk. Full-field measuring techniques have attracted interest in many biomechanical studies, among them, pressure fields on feet [30] or buttocks [31] or strain fields on knee braces [32] or on the calf [33] but, to our knowledge, no direct application on lumbar belts exists. Moreover, it seems that no simultaneous pressure and strain field measurement on a compression device has been reported. In addition, they are strictly non-invasive methods, which is of higher interest for clinical studies. Recently, Bonnaire et al. characterised metrologically pressure mapping systems from FSA© [34] and Molimard et al. presented a new

method for full-field strain measurement by coupling fringe projection and digital image correlation [35]. These methods appeared well adapted for the belt–trunk interface study.

The aim of this paper is to propose and examine a new measurement protocol, based on the aforementioned modern full-field techniques, in order to investigate the mechanical action of lumbar belts in future clinical studies. The variability related to external parameters and the measurement protocol is key to reaching a sufficient statistical power in clinical applications. In a first experimentation, operating conditions are discussed so as to guarantee the best repeatability for pressure and strain for a patient in a static standing position. Then, in a second experimentation, the measurement protocol is used on six different belt configurations as a small illustration of future clinical investigations. At this development stage, only one healthy subject wearing belts is evaluated in order to remove inter-subject variability and to focus on the experimental protocol.

## 2. Materials and Methods

### 2.1. Study Subject and Lumbar Belts

The subject was a 29-year-old male, of bodyweight 81 kg, height 1.81 m and waist circumference 0.96 m with no detected spinal lesion. This subject's freely given and informed consent to participate in this study was obtained. To answer our future objective, i.e., to investigate the mechanical action of lumbar belts, the two measurements performed on this subject wearing a lumbar belt were the measurement of the interface pressure and the 3D strain of the belt.

Three designed types of lumbar belts (Figure 1), each with two back panels (heights 21 and 26 cm), were evaluated, making a total of six conditions. Two of them will be used in the first experimentation establishing the measurement protocol; all the belts will be used in the second experimentation examining the capacity of discrimination of the measurement set-up. These belts are typical of the French market and are prescribed for non-specific chronic low back pain patients. They differ by their structure: Belt B is made of a single textile piece, whereas Belt A and Belt C are an arrangement of elastic bands. Belt A has a lower stiffness than the other two.

### 2.2. Measurements of Pressure and Strain

Both pressure and strain measurement have been extensively described by the authors in [34,35]. Only a brief recall is given here.

Pressure measurements were performed using a matrix of resistive pressure sensors. The action of these sensors is based on the piezoresistive nature of certain materials. The electrical resistivity of these materials changes when a force is applied to the material. Resistivity is proportional to electrical resistance and this is proportional to the voltage, after calibration of the system, and it is therefore possible to obtain the interface pressure by measuring the electrical voltage. These sensors were mounted in a pressure matrix, from which a map of the pressure between the lumbar belt and the trunk was obtained. Fabric creases were avoided by using two matrices for the front and back measurements on the trunk and one matrix for the measurements on each side. These matrices are rectangles 482 by 242 mm$^2$ in size mounted with 384 (12 × 32) sensors and they are calibrated to measure pressures between 0 and 50 mmHg. During the trial, the pressure sensor matrices were attached to a T-shirt with hook-and-loop fasteners on the edges of the matrices above the chest on the T-shirt. Figure 2 illustrates the pressure sensor matrices in position when the measurements are taken.

The 3D strain was measured by coupling a fringe projection method with digital image correlation. Two coordinate systems, global and local, and defined according to the configuration, were used. The global coordinate system ($E_1$, $E_2$, $E_3$) was constructed so that $E_1$ was in the transversal direction, $E_2$ in the antero-posterior direction and $E_3$ in the longitudinal direction. The local coordinate system ($B_1$, $B_2$, $B_3$) was constructed so that $B_1$ was the orthogonal projection of $E_1$ on the plane tangential to the surface, $B_3$ was

perpendicular to the surface at all points and $B_2$ completed the orthogonal base. These frames of reference are illustrated in Figure 3.

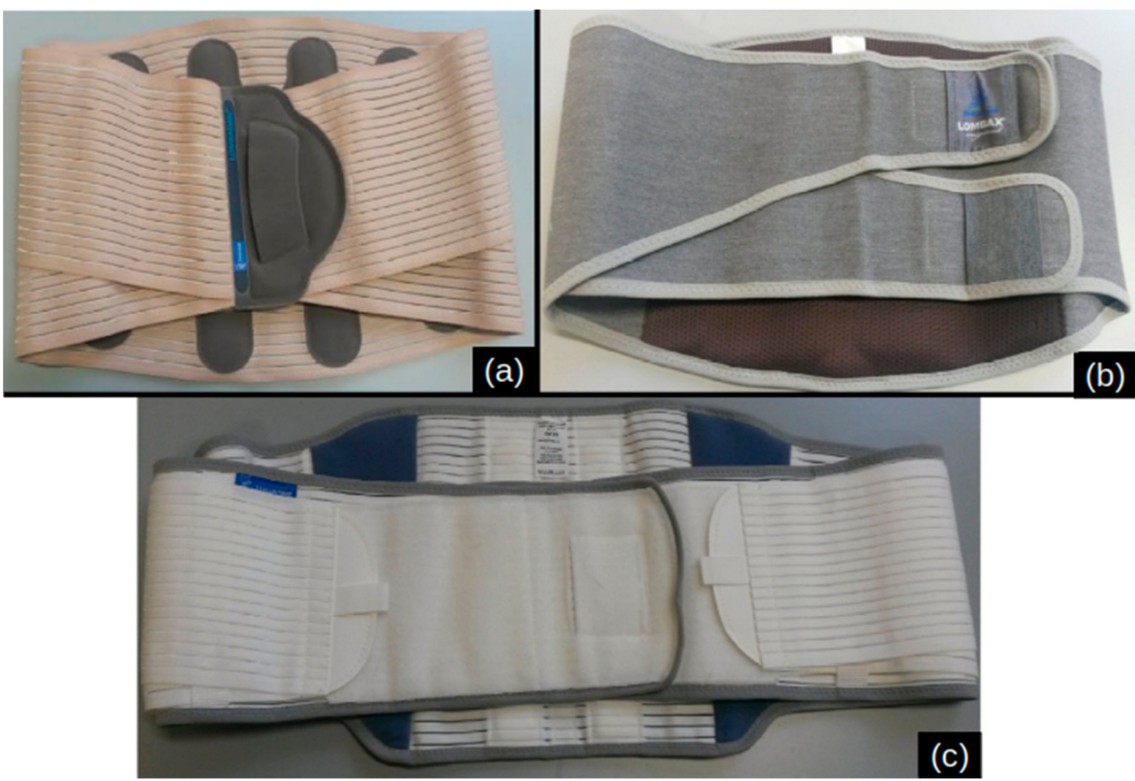

**Figure 1.** Three designed types of lumbar belts studied: (**a**) Belt A (1489 N/m), (**b**) Belt B (2651 N/m), (**c**) Belt C (3244 N/m).

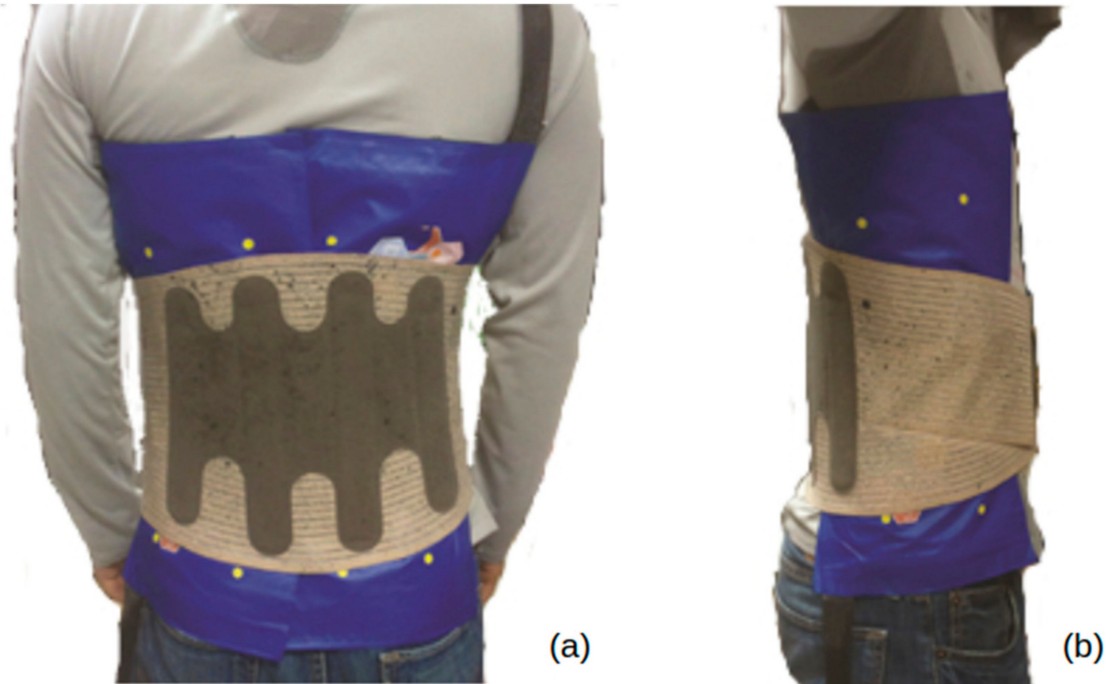

**Figure 2.** Illustration of a pressure sensor matrix positioned on the trunk when measurements are performed (**a**) on the back and (**b**) on the right side.

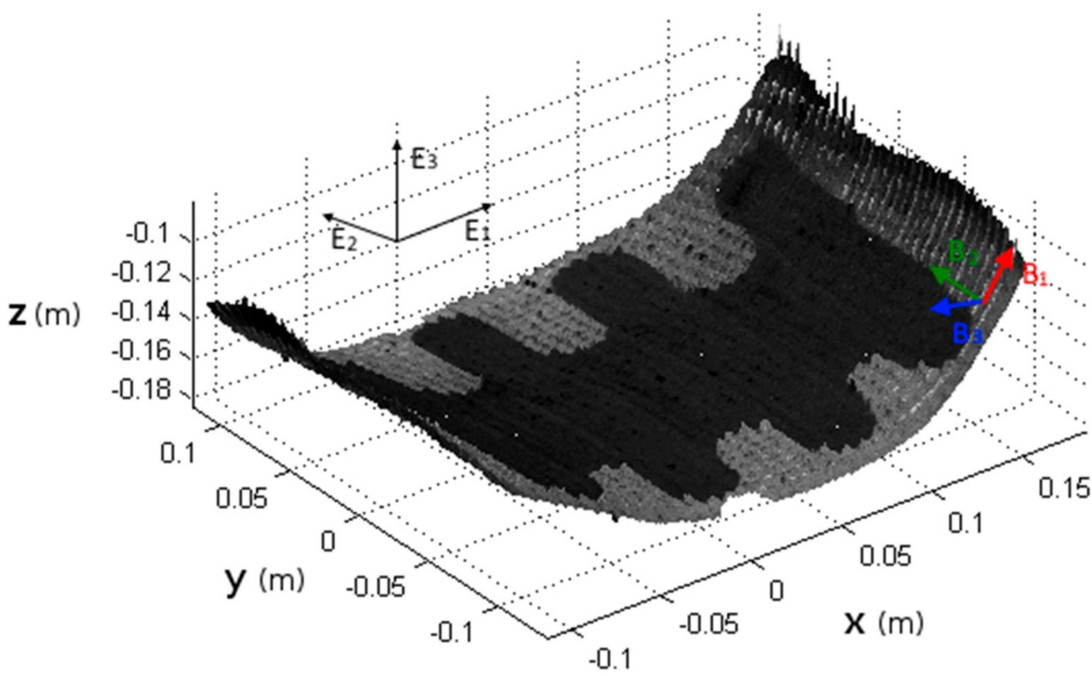

**Figure 3.** Illustration of the shape of the belt on the back in a non-deformed configuration.

The principle of the fringe projection method is to determine the shape of an object by projecting fringes on a reference plane and then on the object under study. Each point of the fringe pattern has a phase, which is modified by the presence of the object. The phase-shift is related to the depth of the object following Equation (2):

$$\varphi(x,y) = S(x,y) \times z(x,y) = \frac{2\pi \times tan\,\theta(x,y)}{p(x,y)} \times z(x,y),\qquad(2)$$

where $z(x, y)$ is the depth of the object, $S$ is the sensitivity of the optical set-up, related to the illumination, to the angle of observation and to the step of the projected grid. A local calibration is necessary to obtain $S$. The phase $\varphi(x,y)$ is calculated by a time-shift method followed by phase unwrapping.

The aim of digital image correlation is to determine the apparent field of displacement between an image $im_0$ described by the grey pattern $f(x, y)$ and a distorted image $im_1$ described by the grey pattern $g(x, y)$, expressed by Equation (3):

$$g(x,y) = T(f(x,y)),\qquad(3)$$

where $T$ is the mechanical transformation between the two images. The transformation parameters ($\delta x$, $\delta y$) are obtained by maximisation of the correlation product ($f \times g$) defined by Equation (4) [35]:

$$h(r,s) = (g \otimes f)(r,s) = \frac{\int_{-\infty}^{+\infty}\int_{-\infty}^{+\infty} g(a,b) \times f(a-r,b-s)dadb}{\int_{-\infty}^{+\infty}\int_{-\infty}^{+\infty} g(a,b)dadb \times \int_{-\infty}^{+\infty}\int_{-\infty}^{+\infty} f(a-r,b-s)dadb}.\qquad(4)$$

The maximum correlation product is obtained in the Fourier transform and the sub-pixel interpolation by extraction of the generalised spatial phase. Digital image correlation requires a speckle pattern, which was produced on the belts with a spray and ink specially designed for marking fabric.

Fringe projection is used to determine the shape of an object; digital image correlation gives the displacement of an object in front of the camera: several steps of data processing are still required to obtain the 3D strain of the lumbar belt. First, the displacements obtained by image correlation are projected onto the surface of the object in order to have the displacement fields in a global frame of reference. Next, the shape is derived to determine the displacement within the local frame of reference of the object. Last, the strain is obtained by the displacement vector gradient in the local frame of reference. Before calculating the gradients, displacement or shape maps were filtered using a salt-and-pepper filter and a gaussian low-pass filter, as prescribed in [35]. The entire measuring process takes approximately 16 s.

### 2.3. Measurement Protocol

The pressure and strain measurements were performed on the static subject in a standing position before and after tightening a given belt. This way, it is possible to measure the mechanical effect of fastening.

The procedure consisted in the following six steps:

1.  the subject puts on the T-shirt,
2.  the pressure sensor matrices are installed,
3.  the belt is put in position, flat against the trunk, but is not tightened,
4.  the shape and the reference images are obtained for the calculation of the displacement,
5.  the belt is tightened by 20% of its length,
6.  the deformed images are obtained for the calculation of the displacement and, at the same time, the interface pressure is measured.

These measurements were recorded on three sides: back, left side and right side of the trunk. No measurements were taken on the front because digital image correlation was not possible on this side: the speckle pattern on the reference image was no longer visible on the deformed image due to the fastening system of the lumbar belt.

### 2.4. Experimentation 1: Evaluation of the Measurement Protocol

The implementation of a measurement protocol requires understanding of all the factors which could influence the results, control factors as well as noise factors and fix, if possible, the latter. Identified possible noise factors that may change the volunteer's abdominal volume or posture in future experiments were:

1.  the T-shirt used to maintain the matrices between the lumbar belts and the trunk, with two different cuts (T-shirt 1 and T-shirt 2);
2.  the breath holding during the measurement, here breath was held at the end of inspiration or at the end of expiration;
3.  the position of the arms during the measurement, here, hands on shoulders, arms crossed or hands on shoulders folded to each side (Figure 4);

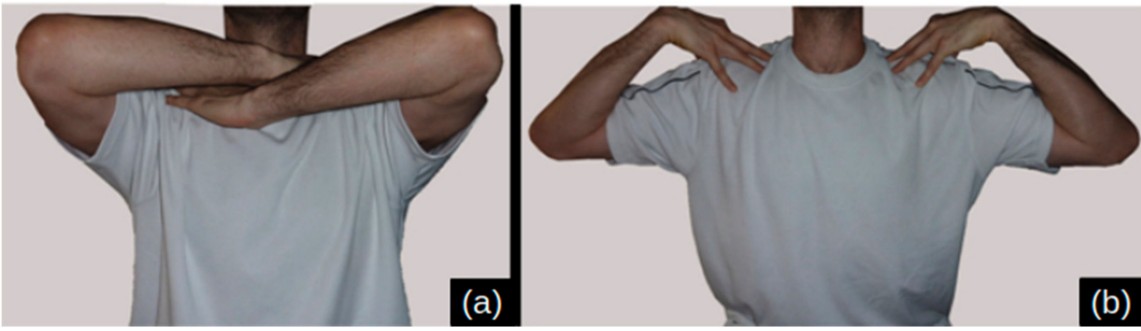

**Figure 4.** Position of the arms: (**a**) crossed on the shoulders or (**b**) folded to each side.

4. In order to ensure that the influence of these parameters, if it exists, was not linked to a specific belt and a height of belt, the influence of the type of belt and its height was also evaluated. Only two belts (Belt A and Belt C) were considered in the study design in order to limit the number of tests. The belts were worn with a 20% overall stretch on the tightening side (right), for all the experiments in this design.

The evaluation of the influence of these five parameters was performed using a factorial study design of $2^{5-1}$ experiments, as detailed Table 1. The output parameters of this study design were the mean of pressures applied by the belt on the trunk and the mean of the strains of this belt for each studied side of the trunk (right, left or back). The chosen experimental design guarantees that main effects can only be confounded with 3-order interactions, but remain independent of each other.

**Table 1.** Experimental design for the analysis of parameters influencing the measurement of interface pressure and strain in the subject.

| Experiment | T-Shirts −1: T-Shirt 1 +1: T-Shirt 2 | Position of the Arms −1: Crossed +1: Folded to Each Side | Breathing −1: End of Inspiration +1: End of Expiration | Belt −1: Belt C +1: Belt A | Height −1: 26 cm +1: 21 cm |
|---|---|---|---|---|---|
| #1 | −1 | −1 | −1 | −1 | +1 |
| #2 | +1 | −1 | −1 | −1 | −1 |
| #3 | −1 | +1 | −1 | −1 | −1 |
| #4 | +1 | +1 | −1 | −1 | +1 |
| #5 | −1 | −1 | +1 | −1 | −1 |
| #6 | +1 | −1 | +1 | −1 | +1 |
| #7 | −1 | +1 | +1 | −1 | +1 |
| #8 | +1 | +1 | +1 | −1 | −1 |
| #9 | −1 | −1 | −1 | +1 | −1 |
| #10 | +1 | −1 | −1 | +1 | +1 |
| #11 | −1 | +1 | −1 | +1 | +1 |
| #12 | +1 | +1 | −1 | +1 | −1 |
| #13 | −1 | −1 | +1 | +1 | +1 |
| #14 | +1 | −1 | +1 | +1 | −1 |
| #15 | −1 | +1 | +1 | +1 | −1 |
| #16 | +1 | +1 | +1 | +1 | +1 |

This experimentation also verified the repeatability of the measurements. This was done by taking a series of three measurements standing with the two selected belts. These measurements were made with the breath held after expiration, wearing T-shirt 1 in the study design and with the arms crossed.

### 2.5. Experimentation 2: Application on Different Belts

The objective of this experimentation is to verify that the measurements can be performed with different belts. Pressure and strain were measured for the six belt configurations worn by the subject in a static standing position. Considering the results of experimentation 1, these measurements were taken with the breath held after expiration, T-shirt 1 in the study design and with the arms crossed.

### 2.6. Statistical Analysis

An analysis of variance for the factorial study design was set up in order to determine the influence of the five parameters tested on the strain and the mean pressure for each side of the trunk. Based on the experimental design, a linear model was used as in Equation (5):

$$Y_k = \beta_0 + \sum_{i=1}^{5} \beta_i X_i, \tag{5}$$

where $Y_k$ is the output variable of the experimental design, $\beta_0$ is the mean value of the linear model, $\beta_i$ is the linear coefficient of the model and $X_i$ the input variable. The results

of this experimental design were presented by tracing the linear coefficients for the various input parameters found by this linear model, and statistically significant influence was calculated using a Fisher test.

The repeatability was assessed by calculating of the mean and quartiles of the three strains and the pressure and using repeated ANOVA. All the distributions were compared by Fisher's exact test first to verify whether these distributions were normal. In the case of a normal distribution, the comparison was performed using Student's *t*-test for paired values. If the values were not normally distributed, the Wilcoxon signed rank test was used. The alpha risk was set at less than 5%.

## 3. Results

### 3.1. Experimentation 1: Evaluation of the Measurement Protocol

The linear regression results are summarised Table 2. The input parameters (T-shirt, the position of the arms, the breathing, the type of belt, the height of the belt) shown by an analysis of variance to have a statistically significant influence on the output variables (pressure or one of the three strains on one side) are indicated by stars in the table. Results for the three strain maps and for the pressure are presented separately. For $e_{xx}$, which is the strain directly related to the belt action, the type of belt and its height are the two relevant factors; for $e_{yy}$, the position of the arms becomes an influencing factor along with the previous ones; $e_{xy}$ is only dependent on the height of the belts. The pressure is significantly influenced by all the factors, but the type of belt and the belt height are related to more variance than the others. Last, sensitivity analysis shows that the output parameters are more influenced by control factors than noise factors, with a percentage of variance from 66% to 93% on pressure, from 85% to 98% on $e_{xx}$, from 42% to 92% on $e_{yy}$ and from 12% to 88% on $e_{xy}$. This result will eventually be better with better control of the noise factors.

Example of measurements of pressure and the circumferential strain $e_{xx}$ for Belt A at the back and Belt C on the left side are represented by box-and-whisker plots in Figure 5 for the three repeated measurements. These box-and-whisker plots were constructed with 120 by 360 measuring points for the strain and 12 by 32 measuring points for the pressure readings. They present the mean, standard deviations and quartiles of this distribution.

The other measurements on the other sides and the other strain presented identical results in terms of repeatability. No statistical difference between the three tests could be found (repeated ANOVA, *p* = 0.78). As Figure 5 shows, the repeatability is sufficient if all the input parameters are correctly controlled.

### 3.2. Experimentation 2: Application on Different Belt Configurations

An example of the strain obtained while wearing Belt A with 21 cm height is shown in Figure 6. According to the B frame of reference, $e_{xx}$ is the strain component in the horizontal direction, tangential to the surface, and $e_{yy}$ the strain component in the vertical direction, tangential to the surface; last, $e_{xy}$ is the shear strain. Figure 6 shows that $e_{xx}$, that corresponds to the stretch direction, monotonically increases from the left to the right side of the patient. Whilst no variations in $e_{xx}$ strain are visible on the left and on the back, more pronounced variations are visible on the right side, that concentrates both the minimum and maximum values. These observations are qualitatively the same—but with lower values—in the vertical direction. The shear strain shows a different pattern, with more a heterogeneous field on the left.

At this stage, these observations cannot be generalised, as they only concern one patient; from a mechanical point of view, we can remark that these global strain variations can be explained by the macro-geometry and the structure of the belt. In particular, fitting a developing surface (the belt) onto a non-developing one (the trunk) gives rise to shear effects clearly visible on the left side. At a local level, the axial strain $e_{yy}$ reflects the woven structure of the fabric making up the belt; the presence of two superimposed fabrics at certain places on the belt is visible on the right side as an horizontal line.

In order to compare the pressure applied by the belts and their strain when being tightened, the results are presented as box-and-whisker plots in Figure 7. Again, these box-and-whisker plots are constructed with 120 by 360 measuring points for the strain and 12 by 32 measuring points for the pressure. These results are given for the strain $e_{xx}$ and for the pressure measured in relation to the side of the trunk measured. For the sake of conciseness, other strain components are not given here, as no statistical differences were found between belts.

Even if the prescribed stretch is 20%, $e_{xx}$ strain shows a high variability in the results. First, mean strain on the right side (mean value close to 15%) is higher than on the left side (mean value close to 8.5%) and on the back (mean value close to 1.4%) for all studied belts. Second, the strain distribution is different in all cases with Belt C than with the other two. Comparable trends are found on the pressure maps. More experiments must be performed to statistically confirm these results.

**Table 2.** Linear coefficients obtained after analysis by the experimental design for: a. applied pressure, b. strain $e_{xx}$, c. strain $e_{xy}$ and d. strain $e_{yy}$, for the three sides of the trunk. *p*-value: ° from 0.05 to 0.1, * from 0.01 to 0.05, ** from 0.001 to 0.01, *** <0.001.

| Sensitivity | $p$ | $e_{xx}$ | $e_{xy}$ | $e_{yy}$ |
|---|---|---|---|---|
| **on the Back** | (mmHg) | ($10^{-2}$ m/m) | ($10^{-2}$ m/m) | ($10^{-2}$ m/m) |
| Mean value | 5.395 *** | 8.42 *** | −2.25 *** | 0.03 |
| T-shirt | −0.44 ° | −0.24 | 0.89 ° | −0.35 |
| Arms position | −0.45 ° | −0.78 | 0.77 | −0.56 |
| Breathing | −0.59 * | −0.23 | −0.37 | 0.29 |
| Belt type | −2.68 *** | −6.87 *** | 0.95 ° | 0.23 |
| Belt height | −0.4 | −1.62° | 1.29 * | −0.13 |
| $R^2$ | 0.94 | 0.89 | 0.67 | 0.22 |
| **Sensitivity** | $p$ | $e_{xx}$ | $e_{xy}$ | $e_{yy}$ |
| **on the right side** | (mmHg) | ($10^{-2}$ m/m) | ($10^{-2}$ m/m) | ($10^{-2}$ m/m) |
| Mean value | 9.42 *** | 18.95 *** | −5.44 | 1.04 |
| T-shirt | −1.12 * | −1.21 | −2.75 | −0.76 |
| Arms position | −1.07 * | −2.06 | 2.90 | −2.64 * |
| Breathing | −0.98 ° | 4.09 | 3.87 | −1.96 ° |
| Belt type | −2.35 *** | 7.56 * | −3.37 | −1.86 |
| Belt height | −1.08 * | −8.52 * | −3.37 | 3.34 * |
| $R^2$ | 0.83 | 0.60 | 0.26 | 0.69 |
| **Sensitivity** | $p$ | $e_{xx}$ | $e_{xy}$ | $e_{yy}$ |
| **on the left side** | (mmHg) | ($10^{-2}$ m/m) | ($10^{-2}$ m/m) | ($10^{-2}$ m/m) |
| Mean value | 11.92 *** | 28.18 *** | 2.53 | −7.37 * |
| T-shirt | −1.30 * | −4.88 | −1.84 | 0.96 |
| Arms position | 0.01 | 4.26 | 1.38 | −0.13 |
| Breathing | −0.49 | −5.99 ° | −1.84 | 2.95 |
| Belt type | −4.47 *** | −20.15 *** | −9.87 ° | 8.26 ** |
| Belt height | −2.22 ** | 7.27 ° | −0.28 | 1.45 |
| $R^2$ | 0.92 | 0.82 | 0.43 | 0.59 |

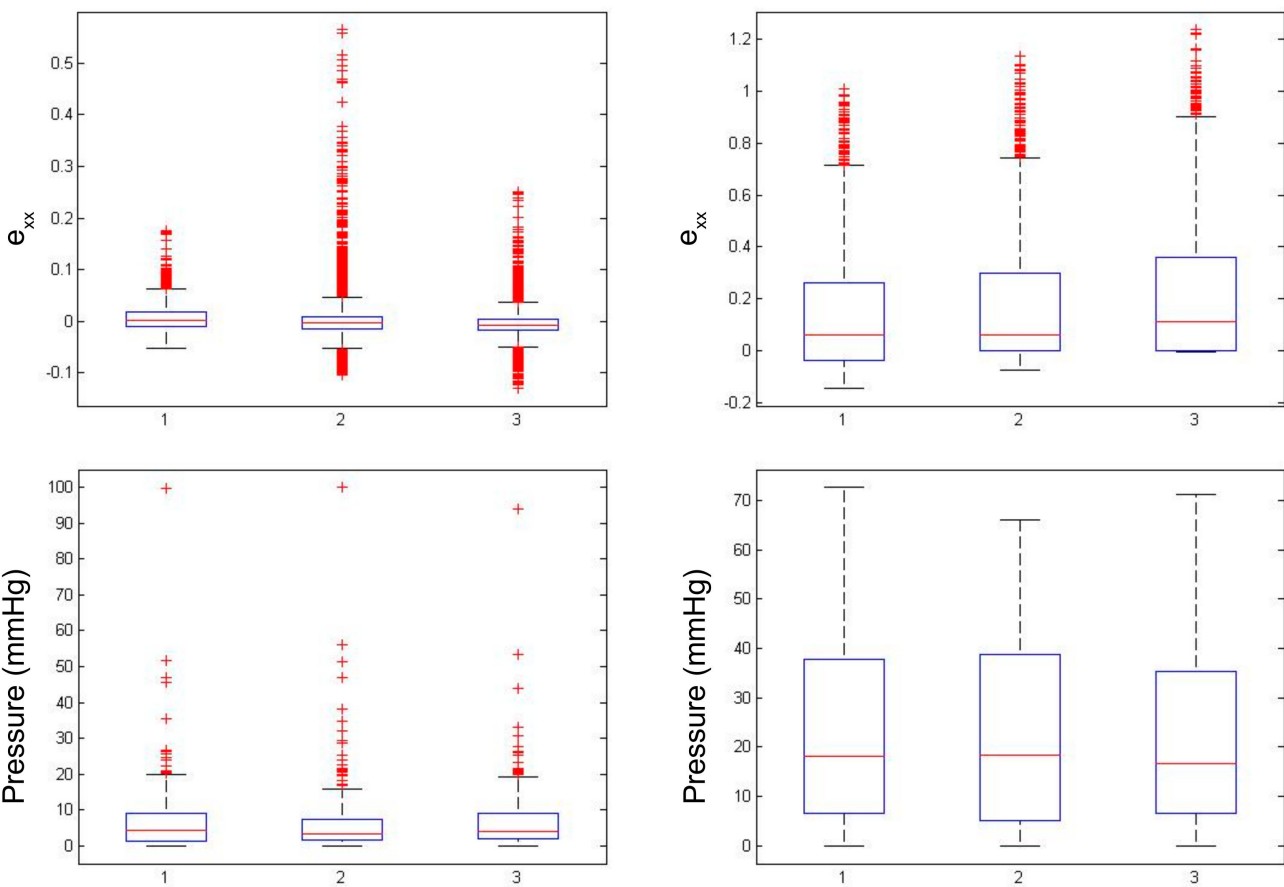

**Figure 5.** Example of the results of measurements for Belt A and Belt C, repeated three times.

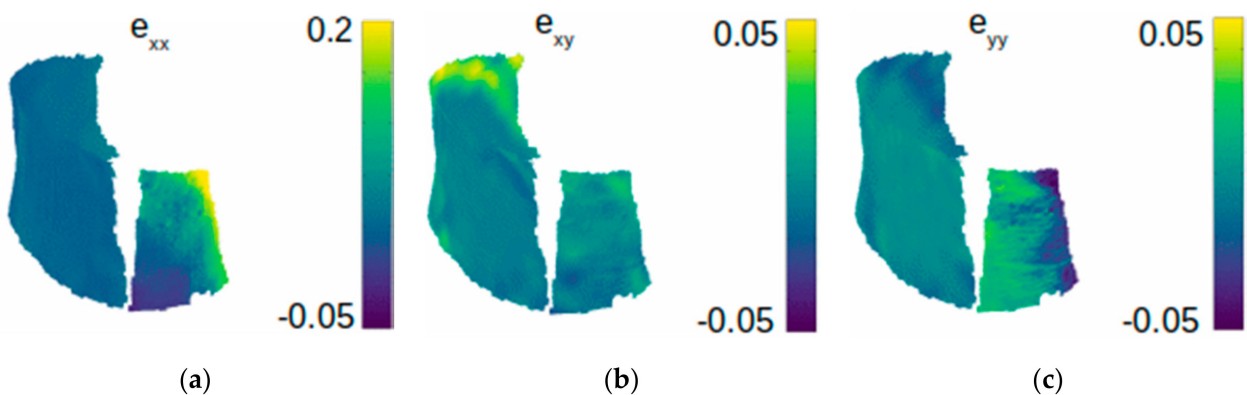

(**a**) (**b**) (**c**)

**Figure 6.** Strain of Belt A of height 26 cm for the various sides measured and the types of strains. (**a**) longitudinal strain; (**b**) transverse strain; (**c**) shear strain.

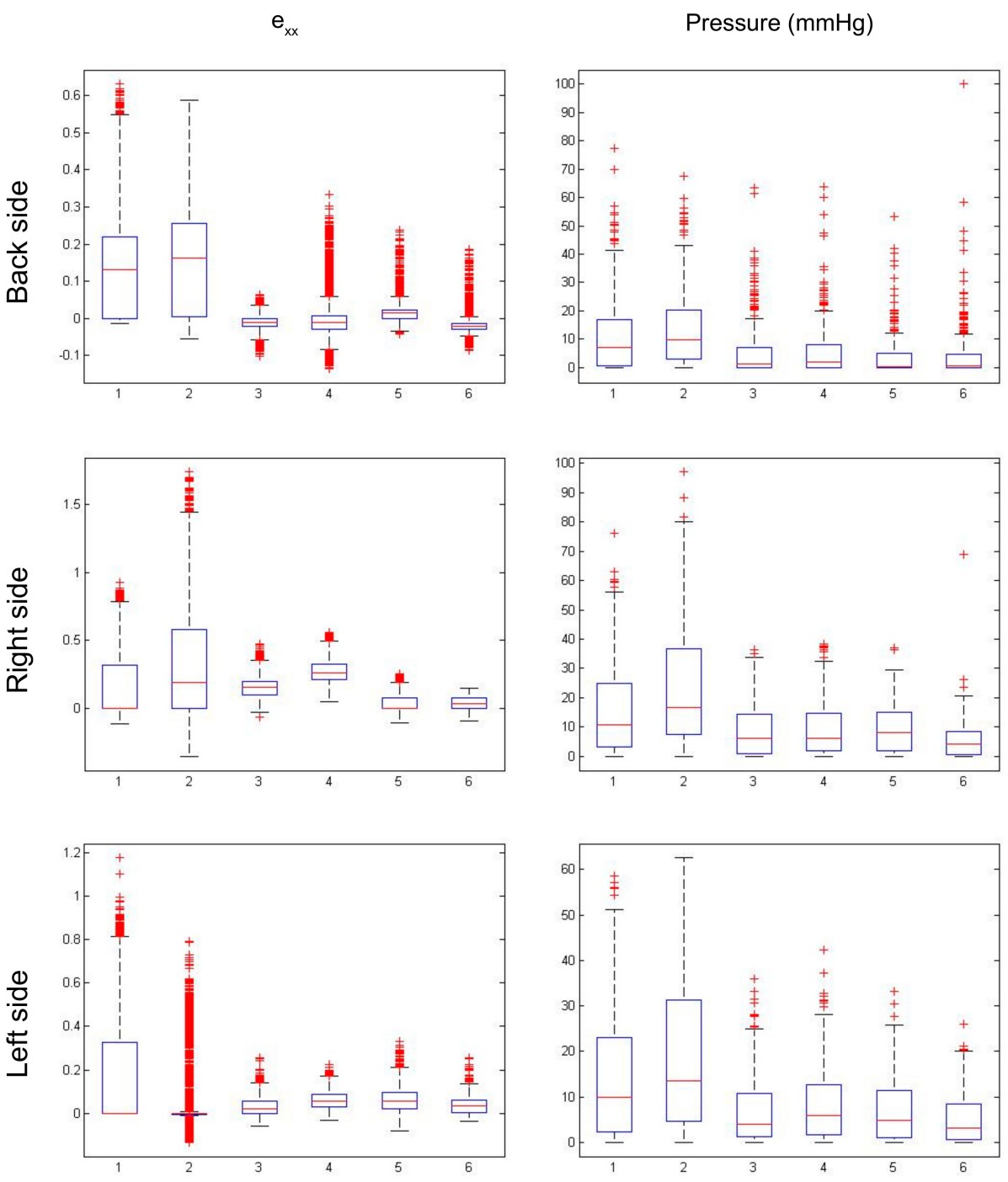

**Figure 7.** Comparison of the belts depending on their strains $e_{xx}$ and the pressure that they exert on the trunk depending on the side measured. Belt numbering: 1: Belt C 21 cm, 2: Belt C 26 cm, 3: Belt A 21 cm, 4: Belt A 26 cm, 5: Belt B 21 cm, 6: Belt B 26 cm.

## 4. Discussion

### 4.1. Experimentation 1: Validity of the Measurements

This experimentation has developed a protocol to measure (i) the interface pressure between the trunk and a lumbar belt and (ii) the strain of the belt when a patient in a static standing position is tightening the belt. These two measurements were validated separately.

The measurement of pressure was validated by a previous study on the metrology of the pressure sensors [34]. Some recommendations were given: the measurements should be made in the same place, within a short time period, by the same operator and for identical surface patterns; the calibration of the pressure matrices should be such that drift of the pressure measurement is avoided; and the pressure matrices should not be superimposed in the measuring zone. All these recommendations were complied with during this experimentation. The same strain measuring device has already been successfully used in another study on knee braces [32] and the same measuring recommendations were applied here.

In addition to being acceptable, the measurement is also repeatable. The three repeated measurements produced identical values for means, quartiles, maxima and minima of strain and pressure as shown in Figure 5; repeated ANOVA concluded that the three measurements could not be distinguished. Nevertheless, this repeatability study was performed on a single subject and should therefore be validated in a larger number of subjects.

The parametric study conducted during the development of this protocol confirms that the parameters having the most influence in the measured interface pressure and strain of the belt are the type of belt and possibly its height. However, this parametric study also revealed that the T-shirt support of the pressure matrices, the method of breath holding during the measurement and the position of the arms have a statistically significant influence on the pressure measurement. It is therefore necessary to be able to compare pressure measurements between the various studies, to fix these three parameters for all subjects and all belts under study. These elements also highlight an interesting fact that the subject's breathing has an influence on the pressure exerted by the belt on the back and that the posture of the person, represented here by the position of the arms, influences the pressure applied on the right side. Breath management as well as the position of the arms are indicators of the importance of movements in the belt's effect on the trunk.

### 4.2. Experimentation 2: Application on Different Belt Configurations

This experimentation compared the mode of action of six belt configurations. On average, there were no significant mechanical differences, whether for strain or the mean pressure exerted for the six studied belts. This may be explained by the fact that all the belts had a similar design, imposed by regulatory affairs. Among other features, they all had four whalebones in the back and one in the front and were constructed mainly of several layers of fabric. Moreover, they were tightened to an identical extent during the experimentation.

For all belts, the $e_{xy}$ strain were a lot smaller than the $e_{xx}$ strain. There was thus little shear on the belts. As this shear can play a part in the comfort of the belts, this result shows that any difference in comfort between the belts cannot be due to shear of these belts.

Nevertheless, Belt C is characteristic in that it has two thicknesses of fabric and has larger pressure intervals. Its pressure distribution is thus different, which may possibly provoke a different internal effect on the trunk. Additionally, the fact that this belt exerts greater pressures than the other belts suggests that these belts could be more efficient and may be less comfortable than the other two. Any difference between the belts therefore depends principally on their technological potential. Belt C's double straps means that more varied pressures can be applied to the trunk and the belt can be adjusted more tightly.

### 4.3. Method of Action of Lumbar Belts

To the authors' knowledge, the protocol designed and presented here is original for fabric medical devices and no other example could be found coupling both pressure and strain measurement. In fact, few examples of full-field strain measurement exist in this domain, and our previous work on knee braces [32] is unusual. Pressure exerted by a compression textile is commonly measured using pointwise measuring systems, for example, in [36–39], and full-field pressure fields could only be found for scoliosis braces [40,41], although a preliminary work has been proposed recently [42].

The measurements obtained with this protocol provide unprecedented information on the mode of action of lumbar belts indicated for non-specific low back pain.

Even though no complete clinical study is presented here and results are obtained for a single subject, some points of interest can be outlined. First, it is worth noting that the measured strain $e_{xx}$ on the right side is identical to the 20% strain applied when the belt is tightened. In contrast, the left side strain of the belt was much smaller; the strain at the back of the belt was even less. The strain of the lumbar belt therefore seems to occur principally on the tightening side. Mean strain in the entire belt is less than the 20% applied on the tightening side. Considering the global body symmetry in the sagittal plane, this might indicate that the Laplace Law is not enough to describe the loading of a belt on the body. In [29], it is suggested that this left/right difference could be explained by the adhesion between the belt and the body, in conjunction with the locking system.

Moreover, the shear strain $e_{xy}$ and the axial strain $e_{yy}$, on all measured sides, were at least ten times less than the circumferential strains $e_{xx}$ on the right side. The principal action achieved by tightening the belt is thus applied in the direction in which it is tightened.

### 4.4. Limitations

Even though this experimental protocol showed convincing results, it is worth mentioning some limitations. First, it aims at giving information on the physical action of lumbar belt on the patient trunk, but low back pain is a complex disease that cannot be only considered from a mechanical point of view. Furthermore, data are external, even though the effect is expected inside the body. This choice is complementary to 3D imaging [43], giving boundary conditions in realistic situations, but requires modeling [12] for an in-depth understanding. Second, because of its technical maturity, this measurement protocol is still limited to clinical research, albeit future applications in clinics could be contemplated. Third, like many others, the 3D strain measuring technique is limited to static situations. The final limitation of this study is that it is based on a single subject and therefore the results obtained for this individual may not apply to others with different anthropometric and physical characteristics. However, we still believe that the results of the study are useful as they provide a clear protocol limiting measuring dispersion.

### 5. Conclusions

The method of action of lumbar belts on the subject trunk still poorly understood, and a new measuring protocol giving a full-field description of the interface pressure as well as of the belt strain is developed and discussed. A pilot study is developed with six different belt configurations on one subject.

The feasibility of the protocol was tested along with its validity and repeatability. The influence of various parameters, which could cause changes in the measurements, such as the choice of the T-shirt for the pressure matrices, breath holding and the position of the arms, was described. Certain recommendations should be implemented to be able to compare studies: the measurements should be in a static standing position; they should be performed with the same type of T-shirt for the pressure matrices; breath holding must use the same technique and the position of the arms must be the same. Within these recommendations, the unexpected variance was sufficiently reduced to achieve good reproducibility of the measurements.

This study showed that full-field measurements were necessary to render the high variability of pressure or strain around the trunk and that the use of the Laplace Law might induce too much simplification.

In the future, this tool will be a means of studying the mode of action of belts for a cohort of patients, and in particular the load transfer to the body that appeared to be more complex than predicted. Such information will help companies to design more efficient belts and medical doctors to prescribe to their patients belts with an appropriate design.

**Author Contributions:** Conceptualisation, R.B., J.M. and R.C.; methodology, R.B. and J.M.; software, J.M.; validation, J.M., R.C. and P.C.; formal analysis, R.B.; investigation, R.B.; resources, J.M. and R.C.; data curation, R.B.; writing—original draft preparation, R.B.; writing—review and editing, J.M., R.C., P.C. and W.S.H.; visualisation, R.B. and J.M.; supervision, J.M., P.C. and R.C.; project

administration, J.M.; funding acquisition, R.C. All authors have read and agreed to the published version of the manuscript.

**Funding:** This research was partially funded by French National Association of Research and Technology under grant 2011/0856.

**Institutional Review Board Statement:** Not applicable.

**Informed Consent Statement:** Informed consent was obtained from the subject involved in the study.

**Data Availability Statement:** Data will be provided to academics by request to J. Molimard (jerome.molimard@mines-stetienne.fr).

**Conflicts of Interest:** R.B. and R.C. are employees of Thuasne. The other authors (J.M., W.S.H., P.C.) declare no conflict of interest.

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
