# Peer review of "Feasibility of a Full-Field Measurements-Based Protocol for the Biomechanical Study of a Lumbar Belt: A Case Study"

_2673-7078, doi:10.3390/biomechanics2020015_

Round 1

Reviewer 1 Report

This is a review of the manuscript “Feasibility of a full-filed measurements based protocol for the biomechanical study of lumbar belt”.

This manuscript describes a renewed instrumentation and measurement protocol for the lumbar brace-trunk mechanical interaction and attempts to validate the methodology. This an interesting study that the reader of Biomechanics would most likely be interested in, especially among those who is working or intended to work in the field of lumbar belt/orthosis design and treatment/rehabilitation for low back pain. However, the current manuscript was not written with clear logic and hard to follow. It could use some extensive editing work to make the main text fluent and easier to understand. More importantly, after reading the manuscript, the reviewer developed some concerns over the quality of this manuscript in its current format, as well as the study design and the way results were presented and interpreted. Some additional explanation would be good.

In general, the authors aimed to measure the interface pressure between the trunk and a lumbar belt and the strain of the belt when tightened. In the reviewer’s opinion, to understand and/or measure such characteristics would require detailed/specific knowledge of mechanics or mechanical engineering. Secondly, as the authors mentioned in the Limitations section that this proposed protocol is still limited to clinical research, what are the connections between the pressure and/or the strain and the clinical outcomes? For example, what can doctors do with the information on the pressure and strain; would these factors influence the outcomes of low back pain treatment or rehabilitation; etc. Therefore, in the Introduction section, the authors should provide an in-depth description of the current status and/or concerns/limitations to build a stronger case for your proposed methodology. Unfortunately, the current Introduction section did not help the reviewer establish the notion that this is an important and urgent issue. Throughout the manuscript, the authors mentioned a couple of times that, “the same strain measuring device has already been successfully used in another study on knee braces”, referring to their prior publication. While the reviewer would agree that this is a “new” application (i.e., being applied to a different joint or body segment) of a previously “validated” methodology, it doesn’t necessarily warrant the plausibility in clinical practices. Body (or joint)-brace mechanical interaction, as the authors referred in the manuscript, should be a function of the joint structure and the corresponding brace design. It involves, at least, the type of joint, degree of freedom, the associated joint actions, and the range(s) of motion. Therefore, the author should also provide some background with respect to the development of measurement protocol and the associated design criteria, such that the audience could have a better understanding when investigating these interactions across other body joints (e.g., ankle, elbow, shoulder). This concern extends to the title of this manuscript, it should be more than “feasibility”, where is the “necessity” and “urgency”.

To be more specific, in the Introduction section, the authors seemed to have paid too much attention to the socio-economic burden of low back pain (LBP), LBP characteristics, as well as the associated treatments, in the first three paragraphs. This information is good to read but may not be relevant to the main scope of the study (i.e., why lumbar belt?). On the contrary, the portion related to the specific of lumbar orthoses/belts was vaguely written. To the reviewer’s knowledge, prescribing the use of lumbar orthose is not a “gold” standard procedure in the treatment of low back pain. Doctors should use their discretion and judgement. As the authors mentioned, in the literature, there has been disagreements regarding the efficacy of lumbar orthoses. What kind of patient characteristics should be considered? Secondly, since there are so many designs, will they make any difference in terms of biomechanical interactions? By addressing these questions would help strengthen the manuscript, especially when the authors stated, “To get into more details, a renewed instrumentation and experimental protocol should bring a better in-sight to the lumbar belt-trunk mechanical interaction.”

Onto the study design and results interpretation, the authors did not provide any details, why these three types of belts were selected. What are the similarities and differences in their design characteristics? Which clinical settings do they apply to, respectively? In the Discussion section, the authors mentioned that the results where no significant mechanical difference was found among the six belt configurations could be explained by the similar design (i.e., whalebones and fabric layers). If this is the case, would it limit the potential of applying the proposed protocol to other belt designs. The reviewer would argue that such results are foreseeable prior to the study design. Therefore, the authors should provide some information on the selection and/or exclusion of belt designs, were they collected from a convenience sample of product or selected based on some criteria? Secondly, the authors listed a total of 5 possible noise factors (i.e., T-shirts, breathing, arm position, belt type, and belt height) and stated that the influence was evaluated using a 25-1 factorial design. Why it was a fractional (i.e.,24) design, not a full factorial (25) design? Which factor was confounded and why? In the Results section, the tables only contained the information of the main effects. Was there any analysis on the potential interactions? It is hard to interpret the results without any information on the interaction terms. With the presence of significant interaction, the interpretation of main effect would be much more complicated.

Some specific questions/suggestions, concerning Table 2, the information can be integrated if the main topic is the significance level. The authors could combine the three tables, by replacing the numeric values with symbols to highlight significant factors across the three parts. Concerning Experimentation 2, first of all, this section was written more like a discussion, with subjective statements like “… shows that the strain is virtually homogeneous …”, “… can be explained by the macro-geometry …”, “… clearly visible on the two sides …”, etc. The reviewer strongly suggests the authors extensively revise the wording in this section to be more objective with solid supporting evidence. The Figure 6 and 7 seemed to be stand alone with no specific descriptions. For example, in Figure 7, in terms of exx and pressure, what was going on with the first two configurations, why they look different from the rest 4 configurations? Thirdly, it seemed that the current data had a lot of potential outliers, would the authors consider any outlier removal algorithm to help “clean” the data before conducting statistical analysis?

Onto the main purpose of this manuscript, the validity of measurement protocol. The reviewer has reservations or concerns over the so-called validation process of a measurement protocol. Throughout the manuscript, the authors suggested that simultaneous measurement of pressure and strain had been successfully conducted, on top of the fact that these measurements had followed recommendations raised by previous studies; hence, the establishment of validity. The reviewer is not an expert in mechanical engineering; therefore, may not have the best knowledge to judge the issue. But in the reviewer’s opinion, it should involve at least some kind of test-retest, or subject-wise comparison process, right? The current study involved only one subject. Would the algorithm work if an obese subject is wearing the belt? What if a patient has impaired mobility with partial trunk range of motion (i.e., can not stand erectly)? Would the proposed protocol work at different trunk postures (e.g., flexion at 70-degree, lateral bending at 15-degree, etc.)? Again, the reviewer is speaking as a general audience, trying to figure out what to do with the proposed protocol.

At this moment, the reviewer decided to bring up major concerns, rather than commenting on the specific issues. The reviewer believes that the comments provided in this report should be addressed or explained by the authors first before moving forward with additional review. Please take your time to prepare your response to clarify all the issues, concerns, and misunderstandings, the process would only help strengthen your work.

Thank you very much!

Author Response

Dear reviewer,

Thank you for the time in reading and commenting our paper. Please find in attachment the detailed response to you comments.

We hope this will suit your expectations, even if this study targeted only feasability for a future clinical study.

Yours

Reviewer 2 Report

This article seeks to establish a protocol to quantify the pressure applied on the trunk from a lumbar orthosis, and the strain in the orthosis itself.  The objective is stated by the authors as not to report the pressure distribution or strain data, but instead to report the validity and repeatability of the data collected.  In addition the authors also outline as an objective reporting the feasibility of this data collection method using a single subject. 

Introduction: The study is first motivated by reporting statistics related to low back pain, including the prevalence and treatment methods.  The authors then introduce lumbar belts as a treatment method, and discuss the difficulty in reviewing any literature due to : heterogeneous populations (no reference is given for this statement), belts from different suppliers (again no references), combining lumbar belts with other treatments, and finally that the detailed mode of action of a lumbar belt has not been outlined in the literature.  However, in the paragraph preceding they state that lumbar belts mechanically reduce pressure on the IVD (lines 54-55), and they have a neuromuscular and proprioceptive effects (line 59).  The authors then argue that with a “renewed instrumentation and experimental protocol” this would “bring a better in-sight to the lumbar belt-trunk mechanical interaction”, which, one assumes would lead to an explanation of the mechanical mode of action.  However, this is not clear, if this is the intent of the study or not, and how the pressure data would inform the mechanical mode of action of a lumbar belt.  The authors should explain this relationship since it is the motivation for the study.  For example, if the authors find a higher pressure distribution at bony prominences, what would that then infer?  Or if they found higher pressure at one side – how would that information be used to infer the mechanical action of the belt?

Lines 74-80.  It has not yet been established how this information (pressure and deformation fields around the trunk, which would vary between subjects, I expect, to a great extent) is “of great importance”.  

Methods:  The methods used should answer the research question (s).  The research question in this case is “to propose and validate a new measurement protocol to investigate the mechanical action of lumber belts”.

The methods then outline a number of variables and a complicated experimental protocol – including three different belt designs each with 2 back panel heights, the effect of 2 different T shirt designs, breathing, and arm position, and it is not clear how all of these variables relate to the research question.  In order to evaluate the measurement protocol, what is the goal of examining results from essentially 6 different belt designs?  As the authors have already simplified by only including one subject – what is the research question being asked with this number of variables related to belt design?

Figure 2 – general comment that these photos are poor quality (grainy)

Line 190: What is the difference between Tshirt 1 and Tshirt 2?

With three measurements in 16 experimental conditions, please defend how you can have statistical power to find these relationships.  The single subject is a limitation of this study and should be noted.  In order to handle the variability created from different subjects, a Repeated measures ANOVA could be done, and this would lend more credibility to your analysis.  If there is a significant difference due to breathing alone, how will this method translate across multiple subjects in much more diverse positions?  It seems that a normalized reference will need to be made, or in other words each subject would have data collected for this type of activity, and then look to see if strain/pressure data were to change to a greater degree when larger movements are part of the experimental protocol.

In short, the objective here is to show what the repeatability of the measures are, so that future studies will be able to interpret a clinically meaningful effect, which would necessarily be larger than the repeatability of the measures obtained in the current study, which are not clinically meaningful.

Author Response

Dear reviewer,

thank you for all your comments. We tried to answer our best in the attached document.

We believe that the objective was not clear enought and made a specific effort on that point. We hope the new version will suit your hopes.

Yours

Reviewer 3 Report

Congratulations for the manuscript " Feasability of a full-field measurements based protocol for the biomechanical study of lumbar belt" is a very original and necessary research as an alternative treatment in back pain.
I have some considerations about the investigation:
First of all, the title I think is not very appropriate. I think it may not be suitable for this research on a single participant.
The summary is not well structured, I miss the structure of the research: Background, Objective, Material and Methods, results and conclusions.
Perhaps it is interesting to emphasize that biomechanical alterations of the kinetic chain of the foot can collaborate in chronic low back pain (for example https://doi.org/10.1177/0309364612471370).
And the most important consideration is that I think it's very important to highlight initial research on a single topic. It is necessary to indicate that future research is necessary.

Author Response

Dear reviewer,

the Scientific Editor suggested answers to your wonders and they have been added to the document. They specifically answer to the following concerns:

  • the title of the document
  • the initial phase of the study and the fact that it only concerns one subject.

Two points are missing, as they are not considered by the academic editor, but I would like to give some answer:

  • this paper in beween experimental mechanics and biomechanics. Therefore, the classical biomechanics structure doesn't suit very well. I do know that it is something questionable but it can be considered easily in journals as "experimental mechanics" or "strain" for example.
  • The topic of the present paper is to measure how a physical treatment (lumbar brace) acts on the trunk. The question of the origin of non specific low back pain is not in our scope, and going in this would deeply change the paper orientation and wouldn't suit with the developments. Maybe this is the reason why academic editor didn't proposed to follow on this point.
    On this topic, we may suggest the reference paper
    Chris Maher, Martin Underwood, Rachelle Buchbinder (2016) Non-specific low back pain, Lancet, 389(10070): 736-747, https://doi.org/10.1016/S0140-6736(16)30970-9

Yours

Round 2

Reviewer 1 Report

The revised manuscript has sufficiently answered and/or addressed all my previous comments and suggestions.

Reviewer 2 Report

The changes to the introduction improve and position the scope of this paper.  As well the inclusion of a model (Laplace) to compare to gives more credibility to the paper, however, this model is never used to validate the results of this paper. As well, I still feel there is little application given that there is only one subject and few if any conclusions can be drawn from this work.  The readership may draw value from this study from the methods used (the measurement protocol), however I think the factors that have been selected would be obvious sources of variation in strain measurement.

No conclusions from this can be made about belt design, so I feel this should be removed from the results.  Conclusions on the mechanism of the belt are also just hypothetical at this stage. 

This study could be much stronger with additional results from additional subjects.
